# Prevalence and Characteristics of Invasive *Staphylococcus argenteus* among Patients with Bacteremia in Hong Kong

**DOI:** 10.3390/microorganisms11102435

**Published:** 2023-09-28

**Authors:** Jonathan H. K. Chen, Hoi-Yi Leung, Charles M. C. Wong, Kwok-Yung Yuen, Vincent C. C. Cheng

**Affiliations:** 1Department of Microbiology, Queen Mary Hospital, Hong Kong SAR, China; lhk755@ha.org.hk (H.-Y.L.); vcccheng@hku.hk (V.C.C.C.); 2Department of Microbiology, School of Clinical Medicine, The University of Hong Kong, Hong Kong SAR, China; wongm29@hku.hk (C.M.C.W.); kyyuen@hku.hk (K.-Y.Y.); 3Infection Control Team, Queen Mary Hospital, Hong Kong West Cluster, Hong Kong SAR, China

**Keywords:** *Staphylococcus argenteus*, *Staphylococcus aureus*, bacteremia, prevalence, whole-genome sequencing, Hong Kong

## Abstract

*Staphylococcus argenteus* is a novel Staphylococcus species derived from *Staphylococcus aureus*. Information on the prevalence and genetic characteristics of invasive *S. argenteus* in Asia is limited. In this study, 275 invasive *S. aureus* complex strains were retrieved from blood culture specimens in Hong Kong and re-analyzed using MALDI-TOF mass spectrometry and an in-house multiplex real-time PCR for *S. argenteus*. The prevalence of invasive *S. argenteus* in Hong Kong was found to be 4.0% (11/275). These strains were primarily susceptible to commonly used antibiotics, except penicillin. Whole-genome sequencing revealed the circulation of three *S. argenteus* genotypes (ST-2250, ST-1223, and ST-2854) in Hong Kong, with ST-2250 and ST-1223 being the predominant genotypes. The local ST-2250 and ST-1223 strains showed close phylogenetic relationships with isolates from mainland China. Antimicrobial-resistant genes (*fosB*, *tet-38*, *mepA*, *blaI*, *blaZ*) could be found in nearly all local *S. argenteus* strains. The ST-1223 and ST-2250 genotypes carried multiple staphylococcal enterotoxin genes that could cause food poisoning and toxic shock syndrome. The CRISPR/Cas locus was observed only in the ST-2250 strains. This study provides the first report on the molecular epidemiology of invasive *S. argenteus* in Hong Kong, and further analysis is needed to understand its transmission reservoir.

## 1. Introduction

*Staphylococcus argenteus* is a novel coagulase-positive Staphylococcus lineage that branched out from classical *Staphylococcus aureus*. This clonal lineage was first reported in Australia in 2006, where it caused skin and soft-tissue infections such as impetigo and necrotizing fasciitis [1,2,3]. It was later proposed to become a new species in 2015 [4]. An increasing number of cases have been reported in different countries across Oceania, Africa, America, Asia, and Europe between 2015–2023 [5,6,7,8,9,10]. Southeast Asia, the Amazon, and remote regions in Australia have shown a high prevalence of *S. argenteus* [11]. Overall, the prevalence of *S. argenteus* among clinical patient specimens in Asia was less than 10% [6,7,12]. However, information about the prevalence and characteristics of invasive *S. argenteus* causing human infections in China and Hong Kong is limited [13,14]. A much higher prevalence of invasive *S. argenteus* was reported in Thailand, with local livestock suspected as a possible reservoir [15]. *S. argenteus* has also been observed in various types of retail foods in the Guangdong province of China [16,17] and Japan, as well as in the environment of slaughterhouses in Japan [18].

*S. argenteus* can cause infections in bone and joints, infective endocarditis, mycotic aneurysm, as well as bloodstream infections [6,7,19,20,21,22,23,24,25]. Cases of food poisoning caused by *S. argenteus* enterotoxins have also been reported in Japan [26,27]. Although the pathogenicity of *S. argenteus* is similar to that of *S. aureus* [11,28,29], higher mortality has been observed among patients with bacteremia caused by *S. argenteus* infections compared to methicillin-sensitive *S. aureus* infections [9]. In an in vitro study, *S. argenteus* demonstrated higher virulence than *S. aureus,* with a 12–15-fold higher expression of the cytotoxic alpha-hemolysin toxin [30].

The two Staphylococcus species exhibit highly similar phenotypic and biochemical characteristics, which makes it challenging to differentiate them using the tube coagulase assay and other automated biochemical identification systems (e.g., VITEK-2) [4]. *S. argenteus*, lacking staphyloxanthin, does not display the characteristic golden color on chocolate agar like *S. aureus.* However, relying solely on physical examination of colony color is subjective, as creamy white colonies have also been reported in *S. aureus* isolates [2,4]. Matrix-assisted laser desorption/ionization time-of-flight mass spectrometry (MALDI-TOF MS) has been proposed as a method to distinguish *S. argenteus* from other members of the *S. aureus* complex. However, species identifications of *S. aureus* complex members are difficult due to their highly similar ribosomal protein structures [31]. In Taiwan, a research group successfully differentiated *S. argenteus* from *S. aureus* by using a self-developed classification model generated through cluster analysis of MALDI-TOF MS results using the ClinProTools software [32].

Besides MALDI-TOF MS, molecular methods targeting specific genes, have been proposed as an alternative approach for identifying *S. argenteus* [31]. Since the overall genome sequences between *S. argenteus* and *S. aureus* shared 90% homology [28], accurate differentiation based solely on analyzing the *16S rRNA* gene sequence is not possible [4]. A conventional PCR assay targeting the 60 amino acid deleted region of a specific hypothetical non-ribosomal peptide synthetase (*NRPS*) gene was used for *S. argenteus* identification [13]. In addition to the *NRPS* gene, the staphylococcal coagulase (*coa*) gene has also been used for molecular typing of the *S. aureus* complex in food products [33,34]. The nuclease (*nuc*) gene has been identified as another potential target for *S. argenteus* identification, as it exhibits a 10% nucleotides divergence or an average nucleotide identity (ANI) of 87% between *S. argenteus* and *S. aureus* [4].

*S. argenteus* belongs to the clonal complex (CC) 75 of *S. aureus*, exhibiting various multi-locus sequence types (MLSTs) [1,3]. According to the PubMLST database, over 60 MLST types of *S. argenteus* have been reported in different geographical regions, with the ST-2250 variant being most frequently reported genotype [11]. Other MLSTs, including ST-1223, ST-1850, ST-2198, ST-2793, ST-2854, and ST-3261, have also been associated with human infections worldwide [8,9,12,21,31,35,36,37,38,39,40]. Different MLST genotypes carry distinct virulence genes and antimicrobial resistance (AMR) genes [40]. AMR genes such as *mecA* and *blaZ* have been identified in 20% or more of the *S. argenteus* strains. Virulence genes encoding enterotoxins and superantigens are commonly found in the chromosome of *S. argenteus*. The presence of CRISPR/Cas loci in *S. argenteus* has been reported to enable the bacteria to capture DNA sequences from phages or plasmids, providing immunity against subsequent phage attacks or plasmid introductions from other staphylococcal strains [40].

In this study, our objective is to determine the prevalence of invasive *S. argenteus* among patients with bacteremia in Hong Kong using MALDI-TOF MS and a self-developed multiplex real-time PCR assay. Furthermore, the genetic characteristics of the identified *S. argenteus* would be further analyzed through whole-genome sequencing (WGS). The identified AMR genes and virulence genes would be compared to those found in other *S. argenteus* strains.

## 2. Materials and Methods

### 2.1. Sample Collection

This retrospective study includes all clinical isolates of the *Staphylococcus aureus* complex that were identified from positive blood culture samples of patients with bacteremia in Queen Mary Hospital, Hong Kong, between January 2020 and September 2021. Queen Mary Hospital is a university-affiliated teaching hospital with 1700 beds located in the Hong Kong West Healthcare Cluster. A total of 275 clinical isolates of *S. aureus* complex, obtained from 275 positive blood culture specimens were retrieved from −80 °C frozen Microbank (ProLab Diagnostics, Richmond Hill, ON, Canada). Prior to further analysis, the isolates were sub-cultured twice on horse blood agar and incubated at 35 °C for 18 h.

### 2.2. MALDI-TOF MS Identification

To confirm species-level identification, each strain within *S. aureus* complex underwent MALDI-TOF MS identification using the MALDI Biotyper SMART system (Bruker Daltonics, Bremen, Germany) with the MBT IVD Library DB-Revision G (10694 MSP). The ethanol formic acid extraction method was used for all 275 isolates. Species identifications with score >2.0 in the top 10 identification results were taken into consideration.

### 2.3. Phenotypic Antimicrobial Resistance Testing

For phenotypic AMR testing, susceptibilities to the following antibiotics were determined using the Kirby–Bauer method: amoxicillin, cefoxitin, ceftaroline, clindamycin, cotrimoxazole, erythromycin, fusidic acid, gentamicin, levofloxacin, minocycline, penicillin G, rifampicin, and vancomycin. The incubation was carried out for 18–24 h at 35 °C in ambient air. The antibiotic susceptibility was determined according to the Clinical and Laboratory Standards Institute (CLSI) guidelines updated in 2023. To ensure quality control, the *S. aureus* ATCC 25923 strain was included in the testing process on each day.

### 2.4. Multiplex Real-Time PCR

Nucleic acid extraction was performed on the 275 culture isolates using the PrepMan Ultra Sample Preparation Reagent (ThermoFisher Scientific, Waltham, MA, USA). The PCR mixture included 5 μL of DNA extract, 1x QuantiNova PCR Probe reaction mix (Qiagen, Hilden, Germany), and the primers and probes listed in Appendix A. An in-house multiplex real-time PCR was conducted on a LightCycler96 system (Roche Diagnostics, Mannheim, Germany), targeting the *S. argenteus* specific *coa* gene, the *S. aureus* specific *nuc* and *sau* gene, and the methicillin-resistant coding *mecA* gene. The PCR conditions were as follows: initial denaturation at 95 °C for 2 min, followed by 45 cycles at 95 °C for 10 s, and 58 °C for 30 s [41,42]. Samples that tested positive for *coa* gene and negative for *nuc*/*sau* gene in the multiplex PCR were identified as *S. argenteus*. Subsequently, their genetic characteristics were further investigated using WGS.

### 2.5. Whole-Genome Sequencing

For the *S. argenteus* strains identified by multiplex PCR, genomic DNA extraction was performed. Briefly, each culture isolate was first cultured in a 3 mL brain heart infusion broth with incubation at 35 °C for 18 h. Genomic DNA of Gram-positive bacteria was extracted using the Qiagen DNeasy Blood and Tissue Kit (Qiagen, Hilden, Germany) with pretreatment lysis specifically designed for Gram-positive bacteria, according to the manufacturer’s instructions. DNA libraries were prepared using the Nextera DNA Prep Kit (illumina Inc., San Diego, CA, USA) and the Nextera DNA CD Indexes (illumina Inc., CA, USA), according to the manufacturer’s instructions. The libraries were sequenced on the MiSeq sequencing system (illumina Inc., CA, USA) using a 2 × 300 bp paired-end read run for 56 h. Prior to genome assembly, the quality of the raw sequencing reads was first evaluated using FastQC (https://github.com/s-andrews/FastQC) (accessed on 15 September 2023) and trimmed with Trimmomatic v0.39 [43]. De novo assembly of the raw sequencing reads was performed using SPAdes v3.11.1 [44], followed by genome polishing using Pilon v1.24 [45]. Prokaryotic gene annotation was performed using Prokka v1.14.6 [46].

### 2.6. Molecular Epidemiology Analysis

The sequences of the seven *S. aureus* housekeeping genes (*arc*, *aroE*, *glpF*, *gmk*, *pta*, *tpi*, and *yqi*) were selected from the Prokka-annotated WGS data using the mlst v2.22.0 (https://github.com/tseemann/mlst) (accessed on 15 September 2023). The MLST of each *S. argenteus* strain was determined using the PubMLST *S. aureus* database (https://pubmlst.org/organisms/staphylococcus-aureus) (accessed on 15 September 2023). The sequence of the *spa* region was extracted using the spaTyper software (https://github.com/HCGB-IGTP/spaTyper) (accessed on 15 September 2023) and matched with the Ridom Spaserver (http://spaserver.ridom.de) (accessed on 15 September 2023). For phylogenetic analysis, the core-genome alignment and variant calls were performed using Snippy (https://github.com/tseemann/snippy) (accessed on 15 September 2023). Genomic regions present in all specimens were selected, and nucleotide substitutions, insertions, or deletions in those core genomic regions were aligned and compared. A maximum likelihood (ML) phylogenetic tree based on single nucleotide polymorphisms (SNPs) in the core-genome alignment of the 11 local strains and NCBI RefSeq sequences of 24 *S. argenteus* isolates collected from different countries was constructed using IQ-TREE 2 [47] and visualized using FigTree v1.4.4 (https://github.com/rambaut/figtree) (accessed on 15 September 2023).

### 2.7. AMR and Virulence Factor Prediction

To determine the presence of AMR-related genes, the assembled contigs were analyzed using the NCBI Antimicrobial Resistance Gene Finder Plus (AMRFinderPlus) v 3.11.18 (Database version 2023-08-08.2) [48]. AMR-related genes with ≥90.0% identity were considered. The presence of virulence-factors-related genes was determined by submitting the assembled contigs to the Virulence Factor Database (VFDB) [49]. The CRISPR–Cas genes in the assembled genomes were identified and subtyped using the CRISPRCasFinder [50]. The CRISPR array size of flanking regions was set to 100 base pairs. The presence of different toxins, adhesins, host defense modulators, and CRISPR/Cas locus was compared among ST types.

## 3. Results

### 3.1. Prevalence of Invasive S. argenteus in Blood Culture Specimens

Out of the 275 *S. aureus* complex isolates obtained from blood culture specimens of patients with bacteremia between January 2020 and September 2021, all were successfully tested using the multiplex real-time PCR. Among these isolates, 11 strains (4.0%) were identified as *S. argenteus*, while the remaining 264 strains (96.0%) were confirmed as *S. aureus* based on their PCR results specifically being *nuc*/*sau*-gene-positive and *coa*-gene-negative) (Table 1). Among the 11 *S. argenteus* strains, none were found to possess the *mecA* gene. In contrast, 132 of the *S. aureus* strains (50.0%) were found to be *mecA* positive.

### 3.2. MALDI-TOF MS Identification for S. argenteus Strains

Among the 11 *S. argenteus* strains, identification with a score > 2.0 was obtained for all strains using MALDI-TOF MS. However, the Bruker IVD MALDI Biotyper was unable to differentiate *S. argenteus* from *S. aureus* or *S. schweitzeri* in all samples, as more than one staphylococcus species were found in the top 10 identification results of the strains with a score greater than 2.0 (Table 1). According to the manufacturer’s guidelines, further investigation would be necessary for species identification.

### 3.3. Genotypes of Invasive S. argenteus in Hong Kong

Based on the assembled WGS data, the 11 local *S. argenteus* strains were classified into three sequence types (ST): ST-2250 (five isolates, 45.5%), ST-1223 (four isolates, 36.4%), and ST-2854 (two isolates, 18.2%) (Table 2). Among the ST-2250 isolates, *spa* types t5078 (two isolates), t7960 (one isolate), t17928 (one isolate), and one unassigned type were identified. The ST-1223 genotype was associated with *spa* types t5142, t12782, and two unassigned types. Another two unassigned types were also found among the ST-2854 genotype.

The SNP phylogenetic ML tree confirmed the three major clusters of *S. argenteus*, which were consistent with their MLST genotyping results (ST-1223, ST-2250, and ST-2854) (Figure 1). Within the ST-2250 cluster, the S7 strain showed a phylogenetic relationship with *S. argenteus* reference strains (XNO62 and XNO106) from China, while the S28 strain exhibited a phylogenetic relationship with reference isolates from Thailand (Colony587, Colony588, and Colony592). Similarly, within the ST-2250 cluster, the S137 and S177 strains were closely related to two other *S. argenteus* strains collected from China (2879A1 and 3343).

### 3.4. AMR and Virulence Factors

All the 11 invasive *S. argenteus* strains were found to be phenotypically susceptible to amoxicillin, cefoxitin, ceftraoline, cotrimoxazole, fusidic acid, gentamycin, levofloxacin, minocycline, rifampin, and vancomycin, based on the *S. aureus* CLSI breakpoints (Table 2). Ten *S. argenteus* strains, excluding strain S28, were phenotypically sensitive to clindamycin and erythromycin. However, strain S28 demonstrated resistance to clindamycin and erythromycin with D-zone development. Five out of the 11 *S. argenteus* strains showed phenotypic resistance to penicillin, and all five strains carried the *blaI* gene. All ST-1223 and ST-2250 strains in this study carried the *fosB* and *tet(38)* genes, which confer resistance to fosfomycin and tetracycline, respectively. The S229 strain carried the highest number of AMR genes, including *fosB*, *tet(38)*, *tet(L)*, *mepA*, *aph(3′)-IIIa*, *sat4*, *ant(6)-Ia*, *blaI*, *blaR1*, and *blaZ*. None of the other methicillin-resistance-related genes, such as *mecC*, were observed in any of the *S. argenteus* strains.

The presence of virulence genes in the 11 *S. argenteus* strains is presented in Figure 2. The virulence factors *lukF-PV* and *lukS-PV*, which code for Panton–Valentine leukocidin (PVL), were not found in any of the local *S. argenteus* strains. Among the ST-1223 strains, genes encoding Staphylococcal enterotoxin (SE) such as *seg*, *sei*, *selm*, *seln*, and *selo* were identified in their chromosome, while the *sec* and *selq* genes were only found in the ST-2250 strains. The ESS pathway secretory proteins coding genes (*esxB*, *esxC*, *esxD*) were present in both ST-1223 and ST2854, but absent in ST-2250. The gene encoding leukotoxin D (*lukD*) was only found in ST-2250 and ST-2854, but not in ST-1223. CRISPR/Cas loci were observed in all ST-2250 (5/5) strains, but absent in all ST-1223 (0/4) and ST-2854 (0/2) strains. The type IIIA CRISPR/Cas locus was found in the local ST-2250 strains.

## 4. Discussion

This epidemiology of *S. argenteus* in Hong Kong has never been reported. This study provides the epidemiological update on invasive *S. argenteus* in Hong Kong. From 2020 to 2021, the prevalence of *S. argenteus* among all *S. aureus*-complex-causing bacteremia infections in Hong Kong was 4.0% (11/275). This figure is similar to the reported rates in other Southeast Asian countries such as Thailand (4.1%) [7], Myanmar (4.5%) [51], and Lao PDR (6.3%) [52]. However, it is higher than the rates reported in Japan (1.0%) [6] and eastern China (0.7%) [13]. Additionally, the prevalence of *S. argenteus* in European and Oceanian countries is much lower compared to Hong Kong [19,20,39,53,54,55].

In this study, two identification methods, MALDI-TOF MS and multiplex real-time PCR, were used to differentiate *S. argenteus* from *S. aureus*. Although MALDI-TOF MS has been reported to accurately differentiate *S. argenteus* from *S. aureus*, it requires the assistance of the peak analysis in the Bruker ClinProTools [32]. However, our data indicate that the routine identification method using the Bruker IVD MALDI Biotyper could not distinguish *S. argenteus* from other members of the *S. aureus* complex. Species identification was confounded by the presence of both *S. argenteus* and *S. aureus* identification with a score > 2.0 in the top 10 identification results. The high homology of ribosomal proteins among the members of the *S. aureus* complex is likely the major cause. The Bruker IVD MALDI Biotyper spectra library (10,694 MSP) included only 8 reference mass spectra for *S. argenteus*. The insufficient number of *S. argenteus* reference spectra in the database could also contribute to this issue. While new *S. argenteus* reference spectra have been added to the spectra library during the recent routine update by Bruker Daltonics, the performance of identification was not evaluated. Another approach to identifying *S. argenteus* would be the use of a self-developed classification model in Bruker ClinProTools. However, developing such a classification model requires extensive resources and expertise, making it unaffordable for most clinical laboratories worldwide. Our data demonstrate that MALDI-TOF MS alone is inadequate for identifying *S. argenteus* in routine clinical diagnosis. Further identification methods are recommended, and molecular tests can be used to differentiate *S. argenteus* from other members of the *S. aureus* complex.

The multiplex real-time PCR used in this study, which targeted the *S. aureus*-specific *nuc*/*sau* gene, *S. argenteus*-specific *coa* gene, and the methicillin-resistant *mecA* gene, successfully identified *S. argenteus* from the *S. aureus*. Based on the WGS data of the *S. argenteus* strains, the primers and probe targeting the *S. argenteus coa* gene demonstrated high specificity. This multiplex PCR assay can be used in clinical laboratory for *S. argenteus* identification. Further analysis will be needed to evaluate the sensitivity of this assay on direct clinical specimens. The 50% positive rate of *mecA* in the *S. aureus* isolates indicates a high prevalence of methicillin-resistant *S. aureus* (MRSA) (50.0%) among bacteremic patients in Hong Kong. On the other hand, methicillin-resistant *S. argenteus*-causing bacteremia was rare in Hong Kong. The low prevalence of methicillin-resistant *S. argenteus* in Hong Kong is consistent with previous reports on *S. argenteus* in Asia (<3%) [16,38,51,56]. *S. argenteus* strains carrying the *mecA* gene are more commonly found in Europe, Australia, and America (13–98%) [31,39,40]. The *mecC* gene is rarely found in the local *S. argenteus* strains and is mainly reported in MRSA circulating in Europe, with its occurrence being rare in Asia [57,58]. Therefore, routine screening of *mecC* gene in *S. aureus* or *S. argenteus* may not be necessary in Asia at this moment.

A high prevalence of AMR genes, including *fosB*, *tet(38)*, and *mepA*, in the chromosome of the Hong Kong strains was revealed. The presence of these AMR genes may hinder the usage of fosfomycin, tetracycline, and tigecycline [59,60,61]. Additionally, the presence of *blaI*, *blaR1*, and *blaZ* genes induce beta-lactamase-mediated penicillin resistance in most of the ST2250 strains in Hong Kong [62]. The phenotypic penicillin resistance of these strains supports our genotypic data findings (Table 2).

Based on our WGS data, multiple genotypes of *S. argenteus* were found to be circulating in Hong Kong and causing bacteremia infections. Both ST-2250 and ST-1223 were confirmed as the predominant genotypes in Hong Kong. Additionally, the ST-2250 strain was also identified in other Asian cities, where it caused various forms of infections [6,8,20,21,24]. The ST-1223 is another emerging genotype that has associations with human infections. Although this genotype had previously been linked to cases of food poisoning, our data confirms that ST-1223 can also be a cause of bacteremia or other invasive infections in humans. Phylogenetic analysis has revealed close relationships among the local ST-2250 and ST-1223 strains in Hong Kong and mainland China. This may be attributed to frequent travel between populations in Hong Kong and mainland China. The presence of *S. argenteus* ST-2250 has been reported not only in patients but also in poultry, raw meat, rice, and flour products in China [16,17]. As Asians frequently come into contact with raw meat and live poultry in wet markets, this may increase the likelihood of *S. argenteus* transmission from retail food to humans, leading to the widespread distribution of *S. argenteus* in Asia.

Compared to the *S. argenteus* strains in Europe and America, a lower number of virulence factors were found in the Hong Kong strains. Key virulence factors such as PVL and toxic shock syndrome toxin-1 (TSST-1) were absent in the 11 Hong Kong *S. argenteus* strains. This suggests that the *S. argenteus* strains in Hong Kong may be less virulent than those circulating in Europe and America. The local ST-1223 and ST-2250 genotypes were found carrying multiple SE genes (*sec*, *seg*, *sei*, *selm*, *seln*, *selo*, *selq*, and *selu*) (Figure 2). These SE genes encode enterotoxins that can cause food poisoning and toxic shock syndrome [63]. The ST-2250 strains in Hong Kong were found to have fewer SE genes and ESS pathway secretory proteins (*esxB*, *esxC*, *esxD*) compared to the ST-1223 strains. The presence of type IIIA CRISPR/Cas locus in the ST-2250 strains only suggests that the ST-2550 strains may actively capture DNA from phage or plasmids sources from other staphylococcal strains, indicating a superior bacterial fitness in comparing to other genotypes. This evolutionary advantage supports the ST-2250 genotype becoming the predominant genotype of *S. argenteus* worldwide.

## 5. Conclusions

*S. argenteus* and *S. aureus* can both cause bacteremia in humans. Due to the differences in antibiogram and genetic characteristics between the two species, proper species identification of *S. argenteus* is important, especially in critical cases such as bacteremia or sterile site infections. For the diagnosis of *S. argenteus*, a molecular assay with multiple PCR targets is recommended instead of MALDI-TOF MS identification. However, MALDI-TOF MS can be utilized if additional reference mass spectra are included in the IVD spectra library in the future. Epidemiological linkage has been identified between the *S. argenteus* strains collected in Hong Kong and mainland China through phylogenetic analysis. Further studies will be necessary to elucidate the transmission reservoir of *S. argenteus* in Hong Kong and other Asian countries.

## Figures and Tables

**Figure 1 microorganisms-11-02435-f001:**
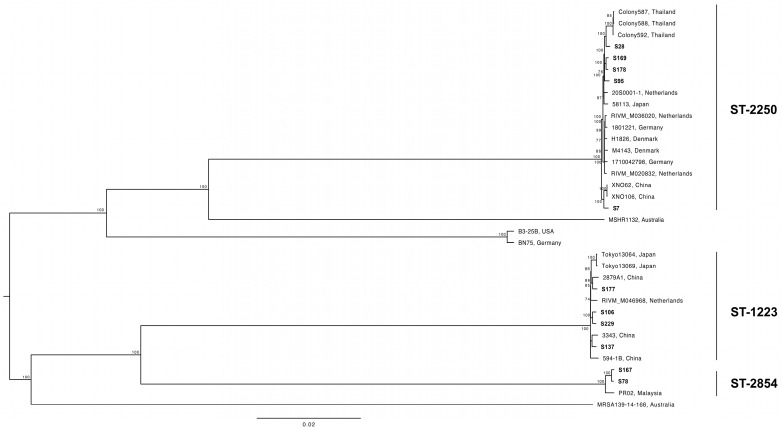
Maximum likelihood phylogenetic tree of 11 local and 24 reference *S. argenteus* sequences based on total core genome SNPs. The tree was constructed by IQ-TREE 2 and was statistically supported by bootstrapping with 1000 replicates. Three MLST clusters (ST-2250, ST-1223, and ST-2854) were identified. The scale bar is in substitution per site. Bootstrap values >70 are shown at the branch nodes.

**Figure 2 microorganisms-11-02435-f002:**
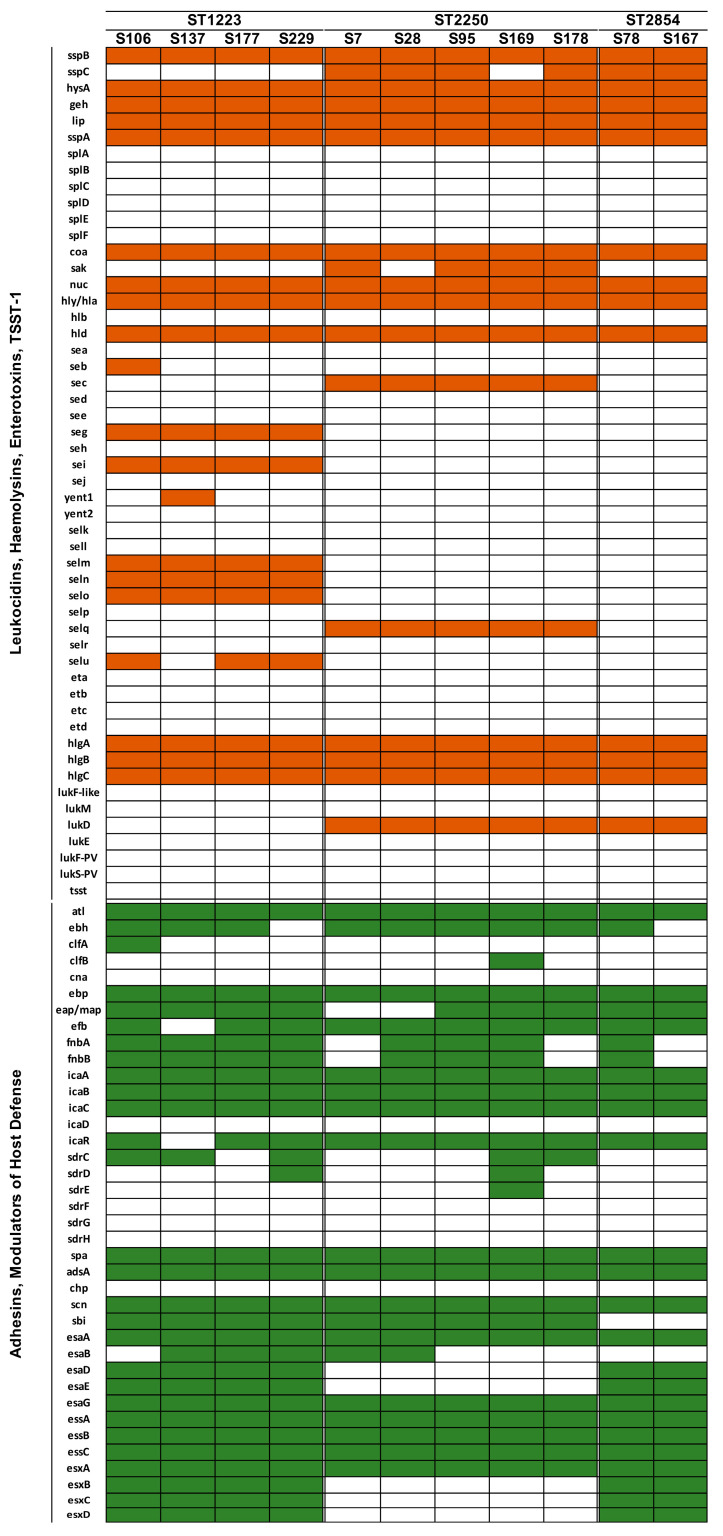
Distribution of virulence factors among the 11 local *S. argenteus* isolates. The coloured bars represent the presence of the factors in the genome of the isolate.

**Table 1 microorganisms-11-02435-t001:** Laboratory identification results of the *S. argenteus* strains isolated in Hong Kong.

Sample No.	Date of Isolation	Gender	Age	Phenotypic and MALDI-TOF MS Identification			Genotypic Identification		
				Gram Stain	Coagulase	MALDI-TOF MS (ID ≥ 2.0)	*nuc*/*sau*	*mecA*	*coa*
S7	29/01/2020	F	53	GPC	P	*S. argenteus*/*S. schweitzeri*/*S. aureus*	N	N	P
S28	09/03/2020	M	84	GPC	P	*S. argenteus*/*S. schweitzeri*/*S. aureus*	N	N	P
S78	13/07/2020	M	62	GPC	P	*S. argenteus*/*S. aureus*	N	N	P
S95	21/08/2020	F	43	GPC	P	*S. argenteus*/*S. schweitzeri*/*S. aureus*	N	N	P
S106	12/09/2020	M	84	GPC	P	*S. argenteus*/*S. schweitzeri*/*S. aureus*	N	N	P
S137	07/12/2020	M	74	GPC	P	*S. argenteus*/*S. schweitzeri*/*S. aureus*	N	N	P
S167	22/01/2021	M	69	GPC	P	*S. argenteus*/*S. aureus*	N	N	P
S169	23/01/2021	F	72	GPC	P	*S. argenteus*/*S. schweitzeri*/*S. aureus*	N	N	P
S177	03/02/2021	M	59	GPC	P	*S. argenteus*/*S. schweitzeri*/*S. aureus*	N	N	P
S178	05/02/2021	M	50	GPC	P	*S. argenteus*/*S. schweitzeri*	N	N	P
S229	22/06/2021	M	93	GPC	P	*S. argenteus*/*S. schweitzeri*	N	N	P

M, male; F, female; GPC, Gram-positive cocci; N, negative; P, positive.

**Table 2 microorganisms-11-02435-t002:** Phenotypic and genotypic antimicrobial resistance profiles of the 11 invasive *S. argenteus* isolates in Hong Kong.

Strain	MLST	*spa* type	Kirby–Bauer Phenotypic Resistance Test														Antimicrobial-Resistant Genes									
			Amoxicillin and Clavulanate	Cefoxitin	Ceftaroline	Clindamycin	Cotrimoxazole	Erythromycin	Fusidic Acid	D-Zone	Gentamycin	Levofloxacin	Minocycline	Penicillin	Rifampin	Vancomycin	fosB	tet-38	tet-L	mepA	aph(3′)-IIIa	sat4	ant(6)-Ia	blaI	blaR1	blaZ
S7	ST2250	t7960	S	S	S	S	S	S	S	−	S	S	S	S	S	S										
S28	ST2250	New1	S	S	S	R	S	R	S	+	S	S	S	S	S	S										
S78	ST2854	New2	S	S	S	S	S	S	S	−	S	S	S	R	S	S										
S95	ST2250	t5078	S	S	S	S	S	S	S	−	S	S	S	R	S	S										
S106	ST1223	New3	S	S	S	S	S	S	S	−	S	S	S	S	S	S										
S137	ST1223	t5142	S	S	S	S	S	S	S	−	S	S	S	S	S	S										
S167	ST2854	New4	S	S	S	S	S	S	S	−	S	S	S	R	S	S										
S169	ST2250	t17928	S	S	S	S	S	S	S	−	S	S	S	R	S	S										
S177	ST1223	New5	S	S	S	S	S	S	S	−	S	S	S	S	S	S										
S178	ST2250	t5078	S	S	S	S	S	S	S	−	S	S	S	R	S	S										
S229	ST1223	t12782	S	S	S	S	S	S	S	−	S	S	S	R	S	S										

The *spa* repeat patterns for the new *spa* types including New1 (299-20-31-25-17-17-16-16-16-16), New2 (299-31-25-22-17-17-17-16), New3 (259-25-17-16-16-16-16), New4 (299-31-25-22-22-43-17-17-16), and New5 (259-25-17-17-16-16-16-16). The presence of specific antibiotic resistant gene is indicated by the highlighted bar.

## Data Availability

The sequence data presented in the study are deposited in the NCBI database under BioProject PRJNA1012691 and WGS accession numbers: JAVKRK000000000 (S07), JAVKRL000000000 (S028), JAVKRM000000000 (S078), JAVKRN000000000 (S095), JAVKRO000000000 (S106), JAVKRP000000000 (S137), JAVKRQ000000000 (S167), JAVKRR000000000 (S169), JAVKRS000000000 (S177), JAVKRT000000000 (S178), and JAVKRU000000000 (S229).

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
