# Peer review of "Prevalence and Characteristics of Invasive Staphylococcus argenteus among Patients with Bacteremia in Hong Kong"

_microorganisms, 2023, doi:10.3390/microorganisms11102435_

Round 1
Reviewer 1 Report
This research contributes to the understanding of the prevalence, genetic diversity, antimicrobial resistance, and potential public health implications of invasive S. argenteus in Hong Kong, shedding light on a previously underexplored aspect of bacterial epidemiology.
The methodology provided in used in this work appears to be generally appropriate for addressing the research objectives, which include determining the prevalence of invasive S. argenteus, analyzing its genetic characteristics, and assessing antimicrobial resistance and virulence factors.
The results and discussion of the study are clear and in agreement with the aims of the study. This fact suggests that the research successfully achieved its objectives and addressed the research questions laid out in the study's aims.
The presented conclusions of the study align with and address the aims set out at the beginning of the research, demonstrating that the study successfully achieved its intended objectives and answered its research questions.
Author Response
Thank you very much for the positive comments.
Reviewer 2 Report
The manuscript titled “The prevalence and characteristics of invasive Staphylococcus argenteus among patients with bacteremia in Hong Kong” goes into great detail to describe findings of this fairly newly discovered bacteria that is closely related to S. aureus. The authors do a good job of bringing forth the diagnostic challenges related to species within the S. aureus complex and use sound methodology to address these challenges. The manuscript is an interesting read and covers an important topic. It does, however, require quite extensive language revision. I have a few specific comments I would like the authors to address.
Line 54-70: Please use passive form when referencing previous studies. If presenting own results, then move this to the discussion/result-section.
Line 67-70: is this an opinnion? No reference is given. Also, which manufacturer?
Line 72: was should be is, correct? The same on line 80. Also see other places in the introduction.
Line 120-124: Please add information on which strain was used for quality control.
Line 123: I do not have the newest version the CLSI standard, but I think oxacillin should be incubated for 24 hours. This may have led to incorrect classification. This should be discussed. Fortunately, mecA PCR was also done.
Line 131: Is it possible that any of the isolates would have carried mecC? This should be discussed at least for the S. aureus isolates.
Line 246: S. argenteus should be in italics.
Much of it is written in past-tense or in active form when passive is preferred.
Author Response
Dear Reviewer,
Thank you very much for the valuable comments.
According to your recommendation, the English language has been extensively reviewed.
Attached please see the responses to your comments.
